# Genetically Encoded Ratiometric pH Sensors for the Measurement of Intra- and Extracellular pH and Internalization Rates

**DOI:** 10.3390/bios12050271

**Published:** 2022-04-25

**Authors:** Lennard Karsten, Lukas Goett-Zink, Julian Schmitz, Raimund Hoffrogge, Alexander Grünberger, Tilman Kottke, Kristian M. Müller

**Affiliations:** 1Cellular and Molecular Biotechnology, Faculty of Technology, Bielefeld University, 33615 Bielefeld, Germany; lennard.karsten@uni-bielefeld.de; 2Biophysical Chemistry and Diagnostics, Medical School OWL, Faculty of Chemistry, Bielefeld University, 33615 Bielefeld, Germany; lukas.goett-zink@uni-bielefeld.de (L.G.-Z.); tilman.kottke@uni-bielefeld.de (T.K.); 3Multiscale Bioengineering, Faculty of Technology, Bielefeld University, 33615 Bielefeld, Germany; j.schmitz@uni-bielefeld.de (J.S.); alexander.gruenberger@uni-bielefeld.de (A.G.); 4Center for Biotechnology (CeBiTec), Bielefeld University, 33615 Bielefeld, Germany; 5Cell Culture Technology, Faculty of Technology, Bielefeld University, 33615 Bielefeld, Germany; raimund.hoffrogge@uni-bielefeld.de

**Keywords:** pH sensor, EGF, ratiometric, GFP, FRET, dynamic range, flow cytometry, live-cell imaging, endocytosis, pH-sensitive fluorescent protein

## Abstract

pH-sensitive fluorescent proteins as genetically encoded pH sensors are promising tools for monitoring intra- and extracellular pH. However, there is a lack of ratiometric pH sensors, which offer a good dynamic range and can be purified and applied extracellularly to investigate uptake. In our study, the bright fluorescent protein CoGFP_V0 was C-terminally fused to the ligand epidermal growth factor (EGF) and retained its dual-excitation and dual-emission properties as a purified protein. The tandem fluorescent variants EGF-CoGFP-mTagBFP2 (pK′ = 6.6) and EGF-CoGFP-mCRISPRed (pK′ = 6.1) revealed high dynamic ranges between pH 4.0 and 7.5. Using live-cell fluorescence microscopy, both pH sensor molecules permitted the conversion of fluorescence intensity ratios to detailed intracellular pH maps, which revealed pH gradients within endocytic vesicles. Additionally, extracellular binding of the pH sensors to cells expressing the EGF receptor (EGFR) enabled the tracking of pH shifts inside cultivation chambers of a microfluidic device. Furthermore, the dual-emission properties of EGF-CoGFP-mCRISPRed upon 488 nm excitation make this pH sensor a valuable tool for ratiometric flow cytometry. This high-throughput method allowed for the determination of internalization rates, which represents a promising kinetic parameter for the in vitro characterization of protein–drug conjugates in cancer therapy.

## 1. Introduction

Several pH-sensitive fluorescent proteins have been developed to investigate the cellular process of endocytosis [1,2,3,4]. The ratiometric pHluorin and SypHer with dual-excitation properties represent frequently used mutants of avGFP or cpYFP [5,6]. Less frequently available are ratiometric fluorescent proteins with dual-emission maxima such as deGFP4, which suffers from low quantum yields [7,8,9], or E^2^GFP, which shows Cl^−^-dependent fluorescence quenching [10,11]. Ogoh et al. demonstrated pH-dependent dual-excitation and dual-emission properties for a green fluorescent protein from *Cavernularia obesa* named CoGFP, which shows only 25% sequence identity with the common avGFP [12]. Several mutations were introduced to obtain five monomeric variants with pK_a_ values in the range of 6 to 10. CoGFP_V0 (S129G, C154S, D156G, K204I, N209Y) revealed green fluorescence (ex. 498 nm, em. 507 nm) with outstanding brightness (quantum yield QY = 0.73; extinction coefficient ε = 218,900 M^−1^·cm^−1^) at pH 7, which decreased upon acidification with a concomitant increase in blue fluorescence (ex. 390 nm, em. 458 nm).

Most studies employ genetically encoded, ratiometric pH sensors, which are expressed inside the cell and recruited to the intracellular compartment of choice by fusing to a corresponding endogenous protein [13,14]. Often, tandem fluorescent proteins are applied to increase the dynamic range of ratiometric pH sensors [15,16,17,18]. Recently, the genetic fusion of mCherry and mTFP1 with the lysosomal-associated membrane protein 1 (LAMP1), named FIRE-pHLy, was used to perform lysosomal pH quantification in a high-content plate-based format [19]. FRET-based pH sensors such as YFpH [20] or the pHlameleon series [21], which originated from the Cy11.5 design [22], were also applied for ratiometric measurements. pH-lemon is another FRET-based pH sensor, which was genetically fused to a glycosylphosphatidylinositol anchor to investigate the secretory pathway involving the endosomal–lysosomal system [23]. In addition to genetically encoded biosensors, there is a huge array of biosensors involving synthetic chromophores or fluorescent nanomaterials [24,25]. An underrepresented approach is the investigation of the pH of intracellular dynamic vesicles upon cellular uptake of a fluorescent protein fusion construct from the environmental cell culture medium, like the HIV-1 Tat-E^1^GFP [3]. However, the endocytosis of extracellular proteins upon receptor binding is important to monitor the cellular internalization, an essential feature of e.g., protein–drug conjugates [26,27] or virus-directed enzyme prodrug therapy (VDEPT) [28,29] in cancer therapy.

Jiang et al. expressed EGFP N-terminally fused to the epidermal growth factor (EGF) in *E. coli* and showed active endocytosis in cells expressing EGF receptor (EGFR) using confocal fluorescence microscopy [30]. The EGFR is a validated target in cancer therapy as elevated expression is associated with various cancer types [31]. Protein vehicles equipped with a cytotoxic payload, e.g., antibody–drug conjugates, are part of the current research for cancer therapy [32,33,34,35]. EGF represents a native ligand of the EGFR, which induces receptor dimerization and clathrin-dependent endocytosis upon receptor binding [36]. The hEGF is a small 6.2 kDa protein with three intramolecular disulfide bonds, which are important for its biological function. For the cytosolic expression of the hEGF in *E. coli*, tendencies to form inclusion bodies and degradation by proteases have been reported [37,38]. However, functional EGF was shown to be efficiently expressed as a fusion protein in the cytosol of *E. coli* [39]. Our group expressed the C-terminal fusion construct EGF-mCherry in *E. coli* and verified the intramolecular disulfide bonds by mass spectrometry and high biological activity [40].

The aim of this study was to develop a genetically encoded ratiometric pH-sensitive fluorescent protein, which can be fused to a protein vehicle for cellular targeting. Application of this extraneously administered pH sensor molecule should allow for a visualization of endocytosis and a high-throughput characterization of the internalization velocity.

## 2. Materials and Methods

### 2.1. Plasmid Construction of CoGFP-His_6_

The gene sequence of CoGFP_V0 flanked by the restriction sites *Nde*I and *Not*I was ordered as string synthesis according to Ogoh et al. [12] and inserted into the pET21a vector with an N-terminal His_6_ tag by restriction cloning. The resulting plasmid was named pET21a/CoGFP-V0_His6 (pZMB0583).

### 2.2. Plasmid Construction of EGF-CoGFP-His_6_

The coding sequence of EGF was amplified using the overhang primers 5′–AAAAAGCTAGCATGAACAGCGACAGCGAGTGCCC–3′ and 5′–AGATCCTCCACCAGATCCACCACCCCTCAGCTCCCACCACTTCAGG–3′ and the plasmid pET21a/EGF-mCherry (pZMB0490) as a template [40]. The coding sequence of CoGFP_V0 was amplified using the overhang primers 5′–GGTGGTGGATCTGGTGGAGGATCTAGCATTCCGGAAAATAGCGGTCTGAC–3′ and 5′–AAAACTCGAGTCAGTGGTGGTGGTGGTGGTGCGGTTTTGCAATTGCGGTTTCATGCTG-3′ and the plasmid pZMB0583 as a template. Both amplification products and the corresponding terminal primers were subsequently used for an overlap extension PCR. The final PCR product was cloned into the pET21a vector using the restriction enzymes *Nhe*I and *Xho*I. The resulting plasmid was named pZMB0492, coding for the EGF-CoGFP fusion construct interspersed by a (GGGS)_2_ linker.

### 2.3. Plasmid Construction of EGF-CoGFP-mTagBFP2-His_6_

The coding sequence of EGF-CoGFP was amplified using the overhang primers 5′–ATACATATGAACAGCGACAGCGAG–3′ and 5′–ACAGATCTGAGGGATCCCGGTTTTGCAATTGCGGTTT–3′ and the plasmid pZMB0492 as a template. The coding sequence of mTagBFP2 was amplified using the overhang primers 5′–GGATCCCTCAGATCTGTTAGCAAAGGTGAAGAGCTCATTAAAGAAAACATG–3′ and 5′–AAAAACTCGAGTTAGTGGTGATG–3′ and the plasmid pET21a/mTagBFP2-His_6_ (pZMB0709) as a template. Both amplification products and the corresponding terminal primers were subsequently used for an overlap extension PCR. The final PCR product was cloned into the pET21a vector using the restriction enzymes *Nde*I and *Xho*I. The resulting plasmid was named pET21a/EGF_CoGFP-V0_mTagBFP2_His6 (pZMB0732), coding for the EGF-CoGFP-mTagBFP2 fusion construct containing the five-amino-acid linker GSLRS between both fluorescent proteins leading to a putative FRET pair in accordance with the C5V construct from Koushik et al. as a FRET reference standard [41,42].

### 2.4. Plasmid Construction of EGF-CoGFP-mCRISPRed-His_6_

The coding sequence of EGF-CoGFP was amplified using the overhang primers 5′–ATACATATGAACAGCGACAGCGAG–3′ and 5′–ACAGATCTGAGGGATCCCGGTTTTGCAATTGCGGTTT–3′ and the plasmid pZMB0492 as a template. The plasmid pRSET-SL-III/mCRISPRed (pZMB0721) was kindly provided by Griesbeck et al. [43]. The coding sequence of mCRISPRed was amplified using the overhang primers 5′–GGATCCCTCAGATCTGTGTCTAAGGGCGAAGAGCTGATCAAGGAAAATATGC–3′ and 5′–AAAAACTCGAGTTAGTGGTGATGGTGATGATGCTTGTACAGCTCGTCCATCCCACCACCAAG–3′ and the plasmid pZMB0721 as a template. Both amplification products and the corresponding terminal primers were subsequently used for an overlay extension PCR. The final PCR product was cloned into the pET21a vector using the restriction enzymes *Nde*I and *Xho*I. The resulting plasmid was named pET21a/EGF_CoGFP-V0_mCRISPRed_His6 (pZMB0731), coding for the EGF-CoGFP-mCRISPRed construct containing the five-amino-acid linker GSLRS between both fluorescent proteins.

### 2.5. Prokaryotic Expression and Protein Purification

Plasmids were transformed into chemical competent *E. coli* BL21(DE3) or *E. coli* Origami B(DE3) cells. An overnight culture was inoculated with a picked clone from an LB agar plate and cultivated for 16 h at 37 °C, 160 rpm on an orbital shaker with an amplitude of 16 mm. Then, a protein expression culture in 0.5 L of LB medium containing 100 µg/mL ampicillin was inoculated to an OD_600_ of 0.1 in a 2 L Erlenmeyer shake flask. Cultivation of *E. coli* Origami B(DE3) required further supplementation with 15 µg/mL kanamycin and 12.5 µg/mL tetracycline. These cultures were grown to an OD_600_ of 0.6–0.8, induced with 0.1 mM IPTG and further cultivated at 37 °C or 30 °C for 4 h or at 18 °C for 16 h. Cells were harvested and stored at −20 °C until purification via immobilized metal-ion affinity chromatography (IMAC). For protein purification, cells were thawed and resuspended in equilibration buffer (50 mM Na_2_HPO_4_, 300 mM NaCl, pH 7.4) supplemented with DNaseI and 1 mM phenylmethylsulfonyl fluoride (PMSF). Cell lysis was performed thrice using a French press (SLM Aminco, Urbana, IL, USA) at a pressure of ca. 110 mPa (16,000 psi). Cell debris was separated from soluble proteins by centrifugation at 15,000× *g* for 30 min at 4 °C. The supernatant was applied to a 1 mL Protino Ni-NTA (Macherey-Nagel, Düren, Germany) column for purification. After washing with 15 column volumes (CVs) of equilibration buffer and 15 CVs of equilibration buffer containing 24 mM imidazole, the protein was eluted with 300 mM imidazole. Chromatography was monitored by measuring UV absorption and conductivity. Proteins were rebuffered into PBS (10 mM Na_2_HPO_4_, 1.8 mM KH_2_PO_4_, 137 mM NaCl, 2.7 mM KCl, pH 7.4) using Amicon Ultra-4, MWCO 10 kDa or 30 kDa (UFC8030, Merck Chemicals GmbH, Darmstadt, Germany) and stored at −80 °C. The concentration of EGF-CoGFP-mTagBFP2 was determined via the absorbance of the chromophore of mTagBFP2 using ε_399_ = 56,000 M^−1^·cm^−1^ [44]. A minor overlap with CoGFP_V0 absorbance at 399 nm was neglected. Concentration of EGF-CoGFP-mCRISPRed was determined via the absorbance at 280 nm (ε_280_ = 65,710 M^−1^·cm^−1^) and corrected by the factor 0.63 to compensate for the absorbance of the chromophores. The correction factor was obtained from an experiment, in which the divergent concentrations of the same EGF-CoGFP-mCRISPRed protein sample were determined via a Bradford assay (considered to be the correct concentration) and the absorbance. Protein aliquots were stored at −80 °C and freshly thawed before usage.

### 2.6. NanoLC–ESI-MS/MS

Protein samples were separated by Laemmli SDS-PAGE and visualized with a Coomassie colloidal stain; then, bands were cut, destained by washing with 30% acetonitrile (ACN) in 100 mM ammonium bicarbonate (ABC) buffer, reduced with 100 µL of 10 mM dithiothreitol in ABC buffer for 1 h at 56 °C and 400 rpm shaking, and finally alkylated using 100 µL of 50 mM 2-iodoacetamide (IAA) in ABC buffer for 45 min at 5 °C and 400 rpm. The in-gel digest using trypsin and collection of resulting peptides was performed according to Herrmann et al. [45].

Peptides were separated with an UltiMate 3000 RSLC Dionex system (Thermo Fisher Scientific, Germany). They were desalted on an Acclaim PepMap™ 100 C18 pre-column cartridge and separated on a 25 cm Acclaim™ PepMap™ 100 C18-LC-column (both Thermo Fisher Scientific, Germany). The effective gradient (15 or 40 min) was 4–30% solvent B (80% ACN, 0.1% formic acid (FA)) using 0.1% FA as solvent A with a flow rate of 300 nL/min. Online ESI-Orbitrap mass spectrometry measurements were carried out by a Q Exactive Plus instrument (Thermo Fisher Scientific, Schwerte, Germany) in data-dependent top 10 acquisition mode. The MS scan range was 350–2000 *m*/*z* with a resolution of 70,000 (m/Δm), and the dynamic exclusion time of precursors for MS/MS was set to 5 s. MS/MS ions were scanned with a resolution of 17,500 and fragmented with normalized CE of 28.

Protein identification was performed with Proteome Discoverer 2.4 (Thermo Fisher Scientific, Germany). The amino-acid sequences of EGF-CoGFP-mTagBFP2 and *E. coli* BL21(DE3) (Proteome ID: UP000002032) were used as templates for peptide spectrum matching. For database searches, trypsin was chosen as enzyme with a maximum of two missed cleavage sites. Oxidation of methionine, N-terminal acetylation, and loss of N-terminal methionine were set as variable modifications, while carbamido-methylation of cysteines was set as a static modification. Mass tolerance ranges were set to 10 ppm for MS and 0.02 Da for MS/MS matches. Calculation of peptide false discovery rate (FDR) occurred via the Target Decoy PSM Validator using a target value of 0.01. For protein identifications, an FDR of 0.01 was also chosen in the Protein FDR Validator node.

### 2.7. Maturation Assay of EGF-CoGFP-mCRISPRed

Purified and freshly thawed EGF-CoGFP-mCRISPRed was stored in PBS (500 mM NaCl, 100 mM KCl, 10 mM Na_2_HPO_4_, 10 mM KH_2_PO_4_, pH 7.5) at 4 °C. UV/Vis absorbance spectra (UV-2450, Shimadzu, Duisburg, Germany) and fluorescence spectra (Spectrofluorometer FP-8300, Jasco, Pfungstadt, Germany) with excitation at 460 nm were measured every 24 h at 20 °C of the same protein sample.

### 2.8. 3D Fluorescence Spectroscopy

EGF-CoGFP-mTagBFP2 and EGF-CoGFP-mCRISPRed were diluted in universal buffer (0.2 M HEPES, 0.2 M Bis-Tris, 0.2 M sodium acetate) at pH 7, 6, 5, or 4 [46]. The 3D fluorescence spectra (Spectrofluorometer FP-8300, Jasco, Pfungstadt, Germany) were recorded at 20 °C with a spectral resolution of 2.5 nm. A hole filter with a transmission of 50% was mounted in front of the cuvette to attenuate the excitation light (2.5 nm slit width), thus allowing for a gentle measurement of the sample. The emission was scanned in 0.5 nm steps starting 5 nm above the excitation wavelength. OriginPro 2021 (OriginLab, Friedrichsdorf, Germany) was used to plot the 3D fluorescence spectra with relative fluorescence intensities.

### 2.9. Live-Cell Imaging

The A431 cell line was cultivated in a μ-Slides 8 Well (ibidi, Gräfelfing, Germany) with 2.5 × 10^4^ cells per 1.0 cm^2^ surface area for 16 h at 37 °C in a 5% CO_2_ atmosphere. Cells were washed with RPMI medium, incubated with 100 nM EGF-CoGFP-mCRISPRed (in PBS, pH 7.4) for 6 h or EGF-CoGFP-mTagBFP2 (in PBS, pH 7.4) for 10 min at 37 °C and 5% CO_2_, washed again, and further incubated at 37 °C and 5% CO_2_. Cells were imaged at indicated timepoints starting from the protein incubation using a confocal fluorescence microscope (LSM 780, Zeiss, Oberkochen, Germany). For the EGF-CoGFP-mCRISPRed fluorescence, the objective LCI Plan-Neofluar 63×/1.3 Imm Corr DIC, main beam splitter (MBS) 458, and a laser at 458 nm (4% power, 1.00 Airy unit) with emission detection at 491–526 nm (32-ch GaAsP detector) and 560–758 nm (photomultiplier tube (PMT)) were used. For the EGF-CoGFP-mTagBFP2 fluorescence, the objective Plan-Apochromat 63×/1.40 Oil DIC, MBS 405 and 488, and a laser at 405 nm (2% power, 0.99 Airy unit) and at 488 nm (2% power, 0.84 Airy unit) with emission detection at 411–485 nm (PMT) and 499–553 nm (32-ch GaAsP detector), respectively, were employed. Different objectives were used due to availability. The emission detection was adjusted to 411–486 nm (PMT) and 499–571 nm (32-ch GaAsP detector) in the case of single-wavelength excitation of EGF-CoGFP-mTagBFP2 with a laser at 405 nm (2% power, 1.01 Airy unit). The line sequential scanning mode was used to minimize the effect of lysosomal movement.

### 2.10. Ratiometric Calibration

Protein solutions were freshly prepared by mixing EGF-CoGFP-mCRISPRed (85 µM in PBS, pH 7.4, stock solution) or EGF-CoGFP-mTagBFP2 (40 µM in PBS, pH 7.4, stock solution) 1:50 with universal buffer [46] (0.2 M HEPES, 0.2 M sodium acetate, 0.2 M Bis-Tris) at defined pH values between 3.5 and 7.5. Droplets of protein solution were placed in a μ-Slides 8 Well and imaged with the objective Plan-Apochromat 10×/0.45 using the same corresponding microscope settings as described for the live-cell imaging experiments. The pinhole was manually adjusted to get adequate fluorescence signals. Then, 12 bit images were processed with Fiji ImageJ 1.52p [47] to perform fluorescence ratio analysis. Mean ratio values were plotted against the pH values, and data points were fitted according to Equation (4) (see Section 3.4) using OriginPro software.

### 2.11. Flow Cytometry

A total of 1.2 × 10^5^ A431 cells were seeded in each well of a 24-well plate and incubated for 16 h at 37 °C and 5% CO_2_. Cells were washed and incubated with 100 nM EGF-CoGFP-mCRISPRed in RPMI for 5 min at 37 °C. Reference cells were only incubated in RPMI medium. Cells were washed and kept at 37 °C and 5% CO_2_ until detaching with accutase for 10 min at 37 °C and 5% CO_2_. Timepoints were measured starting from incubation with EGF-CoGFP-mCRISPRed until the flow cytometry of detached cells. Flow cytometry was performed with the S3e cell sorter (BioRAD, Feldkirchen, Germany). A total of 2 × 10^4^ events were measured per well exciting simultaneously with 488 nm and 561 nm lasers and measuring fluorescence in the FL1 (em. 525/30 nm) and FL3 (em. 615/25 nm) channels. The fluorescence intensity ratio FL1/FL3 was calculated for each event, and mean values versus time were fitted with an exponential decay function.

### 2.12. Microfluidic Application

HEK FreeStyle™ 293-F (#R79007, Thermo Fisher Scientific, Germany) suspension cells were transfected with pcDNA6A-EGFR ECD (1–644) (from Mien-Chie Hung; Addgene plasmid #42666) to express the EGFR ectodomain I–IV on the cell surface [48]. Transfection was performed as described in Boschanski et al. [27]. The microfluidic device for single-cell cultivation was fabricated as described previously [49,50,51]. Briefly, the cultivation device consisted of parallel supply channels, which were connected by cultivation chambers allowing for one cell layer only (200 × 200 × 18 µm^3^). The supply channels were twice as high and had a width of 200 µm. Due to steady perfusion of the whole system, constant environmental conditions could be guaranteed throughout the whole cultivation.

Next, 48 h post transfection, 1.5 mL of 293-F cell suspension (3.7 × 10^6^ cells/mL) was incubated with 100 nM EGF-CoGFP-mTagBFP2 or EGF-CoGFP-mCRISPRed for 10 min at 22 °C to label the cell surface with the molecular pH sensor. The microfluidic cultivation device was manually flushed with pH sensor-labeled 293-F suspension cells until cultivation chambers were sufficiently loaded with cells. The pH of fresh HEK TF medium (Sartorius Xell GmbH, Bielefeld, Germany) supplemented with 8 mM glutamine was adjusted from 5.5 to 7.5, and the medium was constantly perfused through the supply channels by syringe pumps (LA-100, Landgraf Laborsysteme HLL GmbH, Langenhagen, Germany) at a flow rate of 5 μL/min. After 1 h of incubation, cell culture media at five different pH values were sequentially exchanged, and cells were analyzed after 20 min of incubation by inverted fluorescence microscopy (Nikon Eclipse Ti2, Nikon, Düsseldorf, Germany) using a 40× objective and the filter cubes (Nikon, Germany) DAPI-5060C (ex. 377/50 nm, DCM 409 nm, em. 447/60 nm), GFP-3035D (ex. 472/30 nm, DCM 495 nm, em. 520/35 nm), and GFP-3035D combined with mKO/mOrange ET emission filter (ex. 472/30 nm, DCM 495 nm, em. 575/40 nm, AHF analysentechnik AG, Germany).

### 2.13. Fluorescence Ratio Analysis

The 12 bit fluorescence images of live-cell imaging or single-cell cultivation experiments were processed using Fiji ImageJ 1.52p [47] according to the following steps: subtraction of the highest gray value taken from an image with reference cells (i.e., correction of autofluorescence); 32 bit conversion; thresholding based on low values as determined by visual inspection (aided by auto-option) and setting lower values to ‘not a number’ (NaN) (i.e., background elimination); dividing images of both fluorescence channels using image calculator.

For the cells in the cultivation chamber of the microfluidic device, only ratios from the cell surface were considered for mean ratios. Therefore, cell membranes were identified by converting corresponding phase-contrast images into binary masks. Mean ratios of all cells within *n* = 9 chambers were calculated. The unpaired two-sided *t*-test with a *p* < 0.05 level of significance was performed using the software R-4.1.1 to estimate statistical significance between pH shifts.

For protein-solution droplets, the mean ratio was calculated for the whole image area. To generate intracellular pH maps, the resulting ratio images were thresholded to the upper and lower values of the calibration curve. Conversion of ratios to pH values was carried out by pixel calculations according to Equation (5).

## 3. Results

### 3.1. CoGFP_V0 Retains Dual-Excitation and Dual-Emission Maxima as C-Terminal Fusion Protein

We intended to develop proteinaceous pH sensors, which can genetically be fused to ligands; thus, we first tested whether components maintained their functionality as fusion proteins. The dual-emission fluorescent protein CoGFP_V0 and the fusion protein with the epidermal growth factor EGF-CoGFP_V0 were recombinantly expressed in *E. coli* and purified (Appendix A). Protein samples were titrated from pH 7.5 to pH 3.0, and the blue (ex. 380 nm, em. 450 nm) and green (ex. 485 nm, em. 520 nm) fluorescence upon acidification was measured using a fluorescence plate reader (Appendix A). The same fluorescent behavior of decreasing green and concomitantly increasing blue fluorescence was observed for both the free CoGFP-V0 and the fused protein to the C-terminus of EGF. Higher tendencies of aggregation were observed for EGF-CoGFP at lower pH, which resulted in a relatively low fluorescence intensity during 3D fluorescence measurements at pH 5 and 4 (Appendix A), indicating that EGF promotes pH-dependent aggregation. However, aggregation effects have not been reported for labeled EGF in cell culture experiments [30,40,50], which is an important requirement to exploit EGF fusion proteins as cellular pH indicators. Thus, we assume that the low total amount of fusion proteins in endosomes, as well as the spatial fixation by bound receptors, minimizes aggregation in the intended use scenario.

Initial live-cell microscopy imaging experiments with A431 cells, which express a high level of EGFR, showed binding and internalization of EGF-CoGFP_V0 after 8 h incubation at 37 °C (Appendix A). This finding indicated that CoGFP_V0 does not impair the biological activity of recombinantly expressed EGF when fused to the EGF C-terminus similar to an mCherry fusion characterized by Feiner et al. [40]. The emission (at 410–459 nm) of internalized CoGFP_V0 upon 405 nm excitation, which is typical for the violet laser used in other instruments, was generally detectable in acidic vesicles, most likely endosomes (pH 5.8–6.5) and lysosomes (pH 3.5–6.0) [3,19,30,51], but the relatively high background fluorescence and autofluorescence of the cells reduced the sensitivity of the pH probe. Furthermore, excitation at 405 nm was not ideal, as fluorescence emission at 461 nm relatively reached only 78% compared to the excitation maximum at 388 nm at pH 4. This reduced the detection of blue fluorescence of acidified CoGFP_V0. Similar issues with cellular autofluorescence occluding the blue emission were reported for the ratiometric pH indicator deGFP4 [7]. Additionally, lower intracellular concentrations are expected for internalized pH sensor molecules compared to overexpressed molecules of transfected cells. Hence, we concluded that a dual (also called ‘tandem’) fluorescent protein might increase the ratiometric properties of EGF-CoGFP_V0 and allow for a spectral modulation according to the fluorescent partner.

### 3.2. Design and Characterization of Tandem Fluorescent Protein Variants of EGF-CoGFP_V0

Two different tandem fluorescent proteins were designed in combination with the fusion protein EGF-CoGFP_V0. The first tandem protein was designed to increase the green/blue emission ratio and comprised the pH-resistant mTagBFP2 (pK_a_ = 2.7) [44] fused at the genetic level to the C-terminus of EGF-CoGFP_V0 (Figure 1A,B). As a test for improved folding and production yield, EGF-CoGFP-mTagBFP2, which contains three disulfide bridges in the ligand EGF, was also expressed in the cytosol of less oxidative *E. coli* Origami B(DE3) (Appendix A). Comparable expression levels to the standard *E. coli* BL21(DE3) were observed (data not shown). SDS-PAGE analysis revealed only 33% purity upon immobilized metal-ion affinity chromatography (IMAC) as a single purification step (Figure 1D). The target protein was visualized by green in-gel fluorescence. The UV/Vis absorbance of both chromophores confirmed the construction of the tandem fluorescent protein (Figure 1E). The endogenous *E. coli* proteins bifunctional polymyxin resistance protein (ArnA), glutamine-fructose-6-phosphate aminotransferase (GlmS), catabolite activator protein (Crp), and ferric uptake regulation protein (Fur) were identified as major impurities via LC–MS/MS (Appendix A). These host proteins are known to be copurified by Ni-NTA and indicate only moderate folding of EGF-CoGFP-mTagBFP2 in *E. coli* [52]. Purity might be improved by using engineered *E. coli* strains [53] or mutating the most challenging cysteine residues C118 and C226, which are exposed on the β-barrel surface (Appendix A, amino-acid numbering according to multiple sequence alignment). A multiple sequence alignment with seven different derivatives of the two known fluorescent proteins eqFP578 and eqFP611 from *Entacmaea quadricolor* points to possible substitutions for all cysteines present in mTagBFP2 or mCRISPRed.

As a second tandem fluorescent protein, the pH-resistant mCRISPRed (pK_a_ = 2.1) [43] was C-terminally fused to EGF-CoGFP_V0 to achieve a green/red fluorescence ratio upon single-wavelength excitation due to the large Stokes shift of mCRISPRed (Figure 1A,C). Good folding of EGF-CoGFP-mCRISPRed upon expression in the cytosol of *E. coli* BL21(DE) resulted in high purity upon IMAC purification (Figure 1D and Appendix A). In-gel fluorescence revealed only minor fluorescent fragments of the tandem fluorescent protein. The UV/Vis absorbance spectrum confirmed that the combined construct contained both chromophores (Figure 1E). Binding specificity toward EGFR remained for EGF fused to a single fluorescent protein, as well as to a tandem fluorescent protein (Appendix A).

### 3.3. EGF-CoGFP-mTagBFP2 Reveals pH-Dependent FRET and EGF-CoGFP-mCRISPRed Shows Green/Red Fluorescence Ratio upon Single-Wavelength Excitation

The spectral overlap *J* of two fluorescent proteins depends on the wavelength (*λ*)-dependent donor fluorescence intensity *f_D_*(*λ*) with the integral normalized to unity intensity (area under the curve $\int_{0}^{\infty }f_{D}\left(\lambda \right)d\lambda =1$) and the wavelength-dependent extinction coefficient of the acceptor *ε_A_*(*λ*) (Equation (1)). The latter was calculated from the absorption spectra (Figure 2A) and the known extinction coefficient *ε_A_*(*λ**_max_*) according to *ε_A_*(*λ*) = [*ε_A_*(*λ**_max_*)/*A*(*λ**_max_*)] × *A*(*λ*) [56].
(1)$$J=\int_{0}^{\infty }f_{D}\left(\lambda \right)\varepsilon_{A}\left(\lambda \right)\lambda^{4}d\lambda \;(\mathrm{in}\;M^{-1}\cdot {\mathrm{cm}}^{-1}\cdot {\mathrm{nm}}^{4}).$$
(2)$$r_{0}=0.02108\cdot \sqrt[{6}]{\kappa^{2}\phi_{D}n^{-4}\int_{0}^{\infty }f_{D}\left(\lambda \right)\varepsilon_{A}\left(\lambda \right)\lambda^{4}d\lambda }\left(\mathrm{in}\;\mathrm{nm}\right).$$

The Förster radius *r*_0_ is the distance at which the Förster resonance energy transfer (FRET) efficiency is equal to 50% (Equation (2)), and parameters were set as follows: interdipole orientation factor *κ*^2^ = 2/3 based on the dynamic average assumption of a random orientation, quantum yield ϕ*_D_* = 0.64 for the donor mTagBFP2 [44], and refractive index of the medium *n* = 1.3333 corresponding to water.
(3)$$FRET\;E=\frac{1}{1+{\left(\frac{r}{r_{0}}\right)}^{6}}.$$

The FRET efficiency is inversely proportional to the sixth power of the distance *r* between donor and acceptor fluorophore (Equation (3)). FRET only occurs at distances of *r* < 10 nm [56]. A plot of the spectral overlap suggested that mTagBFP2 (Appendix A) functions as donor and CoGFP_V0 as acceptor for a FRET pair at pH 7 (Figure 2A). The integral of the spectral overlap revealed a decrease in area upon acidification because of a decrease in extinction coefficient for the acceptor CoGFP_V0 (Figure 2B). There was no spectral overlap at pH 4. Calculated Förster radii *r*_0_ resulted in 5.5, 4.9, and 3.9 nm for the pH values 7, 6, and 5, respectively (Figure 2C). The *r*_0_ at neutral pH of EGF-CoGFP-mTagBFP2 was slightly higher compared to other FRET pairs mTagBFP-sfGFP (*r*_0_ = 4.6) [56], ECFP-EYFP (*r*_0_ = 4.9) [57], and EGFP-mCherry (*r*_0_ = 5.4) [58], but remained in the midrange of commonly used FRET pairs of today as reviewed by Bajar et al. [56].

FRET efficiency can be increased by reducing the distance *r* between donor and acceptor dipoles or by improving the interdipole orientation factor. Attempts to tightly concatenate CoGFP_V0 and mTagBFP2 to a chimeric protein according to Shimozono et al. resulted in poor folding (Appendix A) [22]. Instead, a five-amino-acid linker following the C5V design as the FRET reference standard of Koushik et al. was applied for EGF-CoGFP-mTagBFP2 [41,42]. According to the obtained 3D fluorescence spectra of the FRET pair (Figure 3), the donor mTagBFP2 (Appendix A) and the acceptor CoGFP_V0 (Appendix A) at pH 7, a FRET efficiency of 45% was indirectly determined, corresponding to an average distance of *r* = 5.7 nm (Appendix A). Under the assumption of a pH-independent distance *r*, the FRET efficiency decreased to 30% and 9% at pH 6 and 5, respectively.

The second pH sensor EGF-CoGFP-mCRISPRed comprising the red fluorescent protein with large Stokes shift was analogously designed and showed constant red (590 nm) and a decreasing green (500 nm) fluorescence upon a single-excitation wavelength at 458 nm.

This ratiometric pH sensor was also compatible with 488 nm excitation, which caused less autofluorescence when used on a cellular level. The initial blue fluorescence (ex. 380 nm, em. 460 nm) originating from CoGFP_V0 at pH 4 and 5 was still detectable. mCRISPRed is known for a longer maturation time compared to its progenitors mRuby3, mRuby2, and mRuby1 (Appendix A). EGF-CoGFP-mCRISPRed expressed at 18 °C required a storage time of 10 days at 4 °C to complete maturation before mature chromophores were applied for ratiometric measurements (Appendix A).

### 3.4. Intracellular pH Mapping Using pH Sensors

EGF fusion proteins produced in the cytosol of *E. coli* have been shown to target human squamous carcinoma-derived A431 cells [40] expressing high levels of EGFR [59,60]. To use the pH sensors EGF-CoGFP-mTagBFP2 or EGF-CoGFP-mCRISPRed for intracellular pH mapping, calibration curves to correlate fluorescence ratios with pH values were obtained. Therefore, pH-titrated protein solutions were prepared for each pH sensor. Ratiometric measurements using the confocal microscope (Appendix A) allowed fitting of the plotted data points with a sigmoidal regression curve (Figure 4A,B). Fluorescence ratiometric calibration curves follow a general scheme of a two-side model of chromophore protonation,
(4)$$R=R_{0}\left(\frac{R_{f}+10^{-\left(\mathrm{pH}-{\mathrm{pK}}^{\prime }\right)}}{1+10^{-\left(\mathrm{pH}-{\mathrm{pK}}^{\prime }\right)}}\right),$$
where *R* represents the ratio, *R*_0_ is the ratiometric offset, *R_f_* is the dynamic range, and pK′ is the ratiometric pK [61]. The ratio-to-pH conversion follows the inverted Equation (4).
(5)$$\mathrm{pH}={\mathrm{pK}}^{\prime }+log\left(\frac{1-\frac{R}{R_{0}}}{\frac{R}{R_{0}}-R_{f}}\right).$$

The pH-dependent FRET efficiency, as well as the pH-sensitive blue fluorescence originally from CoGFP_V0, contributed to the high dynamic range of the green (ex. 488 nm, em, 499–553 nm) to blue (ex. 405 nm, em. 411–485 nm) ratio of EGF-CoGFP-mTagBFP2 and increased its sensitivity as a pH sensor. A pK′ of 6.6 and a remarkable 200-fold ratio change were obtained for EGF-CoGFP-mTagBFP2 between pH 4.0 and 7.5. EGF-CoGFP-mCRISPRed revealed a lower pK′ of 6.1 and a 40-fold change in the green (491–526 nm) to red (560 nm longpass) fluorescence emission ratio at 458 nm excitation (Appendix A).

Next, A431 cells were incubated with EGF-CoGFP-mTagBFP2 for 1 h or with EGF-CoGFP-mCRISPRed for 8 h at 37 °C. Confocal microscopy was performed using the same fluorescence tracks as for the calibration curve (Figure 4C,D,G,H). Autofluorescence of cells was corrected by imaging untreated cells (Appendix A). The intracellular distribution of pixel-to-pixel intensity ratios (Figure 4E,I) was converted into corresponding pH values according to Equation (2) (Figure 4F,J). The pH mapping of the pH sensor EGF-CoGFP-mTagBFP2 worked with dual-excitation (405 and 488 nm) and single-excitation wavelengths (405 nm) exploiting the FRET (Appendix A). The resulting pH maps revealed intracellular vesicles, most probably endosomes and lysosomes, with a pH gradient of up to ΔpH = 1.7 for a single vesicle (Appendix A). Tendencies of lower luminal pH values close to the membrane could be observed, as similarly seen on pH maps from Serresi et al. who performed comparable live-cell imaging experiments upon endocytosis of HIV Tat-E^1^GFP [3]. The localization of the vacuolar proton pump V-ATPase in the vesicle membrane as the primary acidifying agent might explain a spatial pH-gradient pattern [19,62].

### 3.5. Time-Dependent Measurement of Cellular Uptake of the pH Sensors Using Flow Cytometry

Internalization is an important feature of immunotherapeutics embracing antibody–drug conjugates in cancer therapy since cytotoxic payload release usually occurs during the endocytic pathway [63]. Cellular uptake of a compound is a complex process, which can involve the association and dissociation constant toward the targeted receptor, the cellular internalization, receptor recycling, and finally lysosomal degradation. Environmental acidification of the receptor–ligand complex occurs during the transport across the endosomes toward the lysosomes. This intracellular shift to lower pH values could be spatially and temporally resolved on a single-cell level by the pH sensor molecules EGF-CoGFP-mTagBFP2 and EGF-CoGFP-mCRISPRed (Figure 4E,I). Ratiometric flow cytometry offers a high-throughput method to measure these pH shifts in live cells on a single-cell level in a quantitative manner for samples of a cell population and enables obtaining kinetics of cellular uptake. Ratio measurements using flow cytometry give the average fluorescence of each event comprising all cellular uptake processes intracellular and on the cell surface. Time-dependent ratio measurements visualize the rate of cellular uptake, which is a valuable parameter to compare different protein vehicles and their ability to trigger endocytosis upon cellular binding.

Adherent A431 cells were incubated with EGF-CoGFP-mCRISPRed at 37 °C. The commonly used excitation wavelength 488 nm of flow cytometry devices could be utilized for time-dependent ratio measurements by detecting green (525/30 nm) and red (615/25 nm) fluorescence (Figure 5A,B). The fluorescence ratio green/red was individually calculated for each event and plotted for all populations after 0.5 to 3.5 h of incubation (Figure 5C). Sample preparation including cell detachment caused the long dead time for the first measurement. The mean ratios of the populations revealed a decreasing ratio green/red over time (Figure 5D). The data points of the first 2.5 h were fitted with a first-order exponential decay function resulting in a rate of *k* = 0.029 min^−1^. This rate might be used as a complex parameter describing the cellular uptake, as well as indirectly indicating the internalization rate of the corresponding compound.

Organic fluorescent dyes are prevalently used for ratiometric flow cytometry [64,65,66,67]. Valkonen et al. exploited expression of pHluorin2 in *Saccharomyces cerevisiae* strains to monitor intracellular pH homeostasis using ratiometric flow cytometry [68]. However, adaptation of the instrumental optics was required to employ the dual-excitation properties at 405 and 488 nm for pHluorin2 [68,69]. In contrast, EGF-CoGFP-mCRISPRed represents an excellent candidate for ratiometric flow cytometry as it operates with the minimal standard instrumentation of a 488 nm laser and the two emission filters for green and red fluorescence.

### 3.6. Molecular pH Sensor as a Tool for Microfluidic Applications

Single-cell cultivation is an expanding research field with progressing development of microfluidic systems and their applications. The demand for on-chip pH measurement has been met with the implementation of microelectrodes [70,71], different field-effect transistors [72], or the addition of pH-sensitive fluorescent dyes to hydrogel-based microarrays [73], which requires redesigning the microfluidic device and potentially interferes with the original flow design while increasing fabrication complexity. In order to circumvent these problems, the molecular pH biosensor can be applied to a microfluidic device combined with ratiometric fluorescence microscopy as a straightforward indicator for pH changes.

A biological system was established with HEK suspension cells transiently expressing the EGFR ectodomain on the cell surface, which could be bound by the EGF ligand of the pH sensor protein [48]. Cells showing a transfection efficiency of ~56% (Appendix A) were incubated with EGF-CoGFP-mTagBFP2 or EGF-CoGFP-mCRISPRed, flushed into monolayer cultivation chambers of the microfluidic system, and cultivated for 3 h at 22 °C (Figure 6A,D,G,J). The fluorescence emissions at 447 nm (ex. 377 nm) and 520 nm (ex. 472 nm) of EGF-CoGFP-mTagBFP2 (Figure 6B,E), or at 520 and 575 nm (both ex. 472 nm) of EGF-CoGFP-mCRISPRed (Figure 6H,K) were detected by fluorescence microscopy after exchanging the various pH-titrated cell culture media inside the microfluidic device. Only the fluorescent signals coming from the cell surface were considered for the ratiometric evaluation (Figure 6C,F,I,L). EGFR–pH sensor complexes might have been internalized because of membrane recycling during the 3 h cultivation and were therefore no longer exposed to the surrounding culture media. In the pH scope of 7.5 to 5.5, EGF-CoGFP-mTagBFP2 and EGF-CoGFP-mCRISPRed showed a significant change of mean ratios for pH steps of ΔpH = 1.0 and 0.5, respectively (Figure 6M,N). This demonstrates the applicability of the pH sensors to track the extracellular pH in a broad range during single-cell cultivation on-chip without further modifications of the microfluidic device.

A different approach to avoid redesigning the microfluidic device is the absorbance measurement of phenol red-containing medium, which is limited to a pH range of 7 to 8 [74]. Fluorescent dyes were also directly added to the culture medium [75]. However, a constant flow of high concentrations in a micromolar range is required, whereas specific targeting to the cell membrane with the biosensor requires only nanomolar concentrations. Generally, the EGF part of the pH sensor molecules can be flexibly exchanged to ligands, which bind to natively expressed antigens on the surface of the cultivated cells [3]. Alternatively, the genetically encoded biosensor might be directly expressed on the mammalian cell surface with a peptidic transmembrane anchor or using a glycosylphosphatidylinositol anchor [14], or on a prokaryotic cell surface with a C-terminal fusion to the ice-nucleation protein [76]. Immobilization of the pH biosensor on microbeads would be conceivable as well to have flowing particles as ratiometric pH indicators [77].

## 4. Conclusions

We demonstrated that the ratiometric pH sensors EGF-CoGFP-mTagBFP2 and EGF-CoGFP-mCRISPRed with high dynamic ranges are excellent indicators for intracellular pH upon EGFR binding and internalization. The moderate protein folding due to cysteine-rich fusion constructs such as EGF-CoGFP-mTagBFP2 shows the importance of fluorescent protein engineering of cysteine-free variants such as cgfTagRFP [78]. Our genetically encoded biosensors allowed for a spatially precise pH mapping in live-cell imaging experiments. Furthermore, monitoring of extracellular pH shifts was demonstrated, which could be applied in single-cell cultivation using fluorescence. Moreover, EGF-CoGFP-mCRISPRed exhibited high relevance for ratiometric flow cytometry as a high-throughput method to semi-quantify time-dependent cellular internalization. The combination of a pH sensor with the native ligand EGF can be used as convertible platform for the evaluation of antibody–drug conjugates. Fusion constructs of fluorescent proteins to different sites of full-length immunoglobulin-G have been successfully expressed [79,80,81]. The simple fusion of the tandem fluorescent protein pH sensor to tumor-targeting antibody candidates would lead to a unique comparable kinetics for internalization. The endocytosis of antibody–drug conjugates is essential for drug release inside the cancer cell and has a huge impact on the tumor regression in xenografted animal models. Further research on this characterization method might improve the prediction for promising antibody candidates on the way from in vitro to in vivo analysis.

## Figures and Tables

**Figure 1 biosensors-12-00271-f001:**
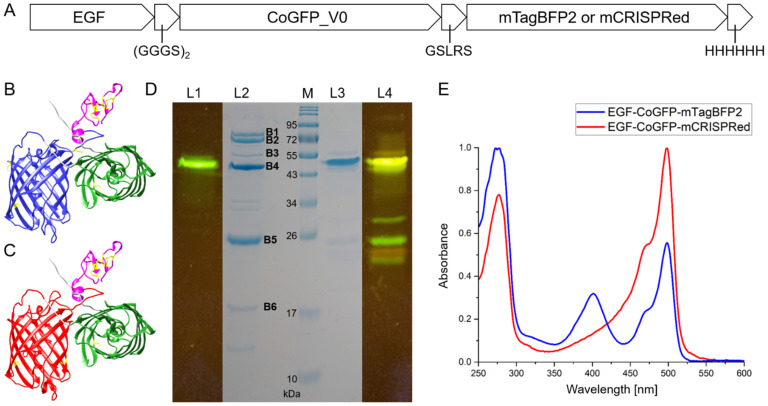
Model and initial characterization of ligand–tandem fluorescent proteins. (**A**) Linear scheme of pH sensor fusion proteins containing EGF, a (GGGS)_2_ linker, CoGFP_V0, a five-amino-acid linker GSLRS, mTagBFP2 or mCRISPRed, and a C-terminal His_6_-tag. (**B**) Structural model of EGF-CoGFP-mTagBFP2 calculated with ColabFold [54]. (**C**) The mCRISPRed structure (PDB: 6XWY) was inserted for EGF-CoGFP-mCRISPRed using UCSF Chimera [55]. Ribbon structures show EGF (magenta), CoGFP_V0 (green), mTagBFP2 (blue), mCRISPRed (red), and all cysteine residues (yellow). (**D**) Reducing SDS-PAGE of EGF-CoGFP-mTagBFP2 (**L1**–**L2**) and EGF-CoGFP-mCRISPRed (**L3**–**L4**) after a single purification step via IMAC. Samples were not heat-denatured to provide in-gel fluorescence upon blue light (470 nm) exposure (**L1**,**L4**) before Coomassie staining (**L2**–**L3**). Contaminating protein bands (**L2**) were identified as endogenous proteins from *E. coli* via LC–MS/MS (band intensities in parentheses): ArnA (**B1**, 10%), GlmS (**B2**, 14%), EGF-CoGFP-mTagBFP2 (**B3** and **B4**, 3 and 33%), Crp (**B5**, 27%), Fur (**B6**, 4%). (**E**) Normalized UV/Vis spectra of EGF-CoGFP-mTagBFP2 (blue) and EGF-CoGFP-mCRISPRed (red) in PBS at pH 7.4.

**Figure 2 biosensors-12-00271-f002:**
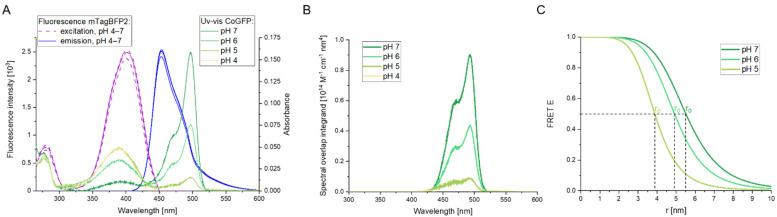
pH-dependent spectral overlap of the FRET donor mTagBFP2 and acceptor CoGFP_V0. (**A**) Excitation (at 455 nm em., purple) and emission (at 400 nm ex., blue) spectra of mTagBFP2 extracted from 3D fluorescence spectra at pH 4 to 7 are plotted together with the UV/Vis absorbance spectra of CoGFP_V0 at pH 4 to 7 (green). The pH-dependent absorbance spectra of CoGFP_V0 overlap with the pH-resistant fluorescence emission of mTagBFP2, indicating a pH-dependent FRET pair. (**B**) The spectral overlap integrand was calculated as product of *f_D_*(*λ*), *ε_A_*(*λ*) and *λ*^4^ according to Equation (1) with mTagBFP2 as donor and CoGFP_V0 as acceptor for the pH values 7, 6, 5, and 4. The integral of the spectral overlap decreased upon acidification. (**C**) FRET efficiencies E were calculated as a function of the distance *r* between donor and acceptor according to Equation (3). FRET E was not calculated for pH 4 as no spectral overlap was observed. The Förster radius *r*_0_ is the distance at which FRET E is equal to 50%. According to Equation (2), *r*_0_ was equal to 5.5, 4.9, and 3.9 nm at pH 7, 6, and 5, respectively, for the FRET pair CoGFP_V0-mTagBFP2 due to a decreasing spectral overlap with lower pH values.

**Figure 3 biosensors-12-00271-f003:**
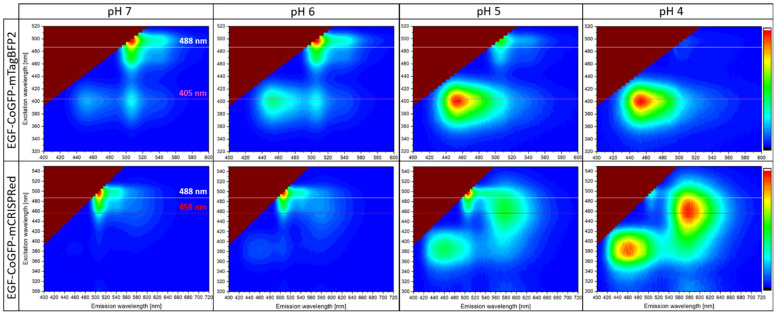
The 3D fluorescence spectra of EGF-CoGFP-mTagBFP2 and EGF-CoGFP-mCRISPRed at pH 7, 6, 5, and 4 were measured using a spectrofluorometer. Common suitable excitation wavelengths 405 nm (purple), 458 nm (red), and 488 nm (white) are marked. Intensities vary significantly and were normalized to the maximum for each plot. Plots were generated using OriginPro.

**Figure 4 biosensors-12-00271-f004:**
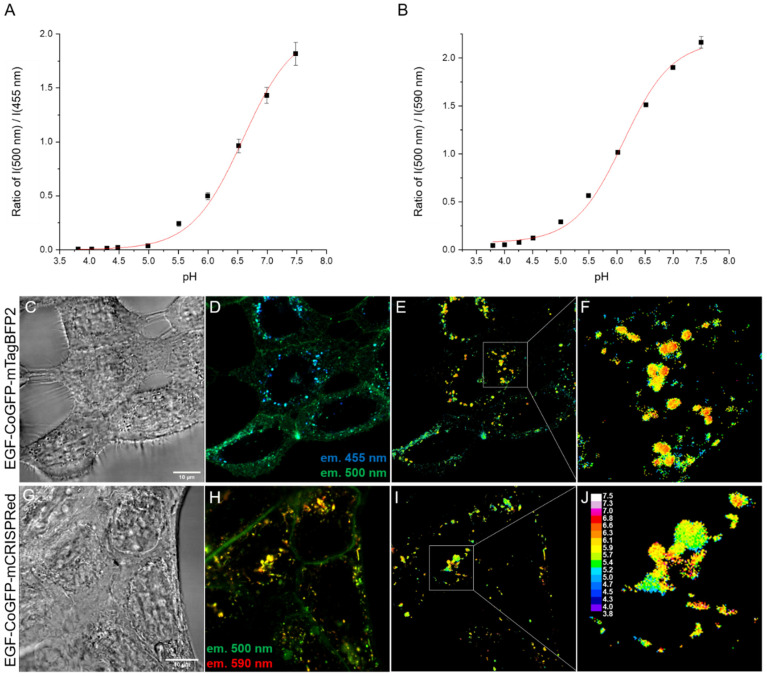
Fluorescence measurements of titrated (**A**) EGF-CoGFP-mTagBFP2 or (**B**) EGF-CoGFP-mCRISPRed protein solutions were performed using confocal microscopy, and mean values of emission ratios of intensities at 500 nm (ex. 488 nm)/455 nm (ex. 405 nm) or 500 nm (ex. 458 nm)/590 nm (ex. 458 nm) were calculated. Mean values of ratios were obtained from processed images using ImageJ software [47]. Data points were fitted according to Equation (1) using OriginPro software. Error bars represent the standard deviation. A431 cells expressing high levels of EGFR were incubated with 100 nM (**C**–**F**) EGF-CoGFP-mTagBFP2 or (**G**–**J**) EGF-CoGFP-mCRISPRed for 10 min, washed, further incubated for (**A**) 1 h or (**B**) 8 h at 37 °C, and imaged using confocal fluorescence microscopy. (**C**,**G**) Scale bars in the lower right corner of the bright-field images represent 10 µm. Pixel-to-pixel intensity ratios (**D**,**H**) were converted with the corresponding calibration curve into pH values (**E**,**I**). The resulting intracellular pH maps were pseudo-colored according to the calibration bar. White boxes indicate selected areas for (**F**–**J**) image magnifications.

**Figure 5 biosensors-12-00271-f005:**
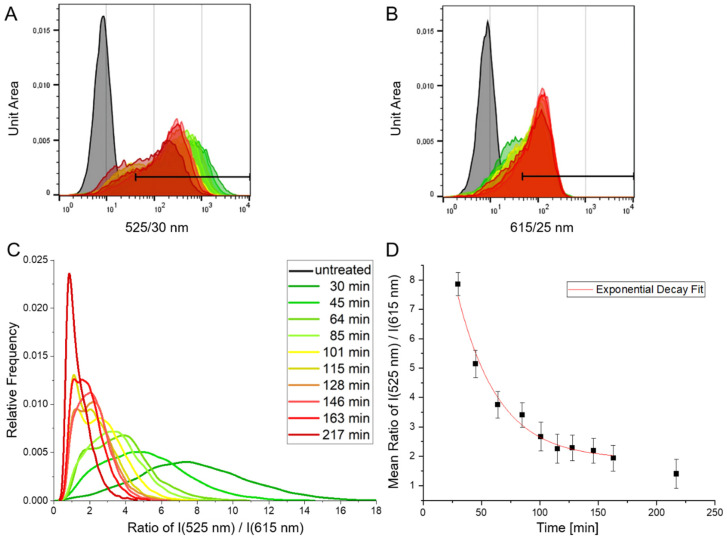
A431 cells expressing a high level of EGFR were incubated with EGF-CoGFP-mCRISPRed at 37 °C, and fluorescence emission was measured at the indicated timepoints using flow cytometry. Histograms show fluorescence emission at (**A**) 525/30 nm (**B**) 615/25 nm for each timepoint upon 488 nm excitation. Bar gates indicate cells with a fluorescence above that of 99% of untreated reference cells. Each measured population comprises 13,000–17,000 events. (**C**) The ratio of emission intensities at 525 nm to 615 nm was calculated for each cell. (**D**) Mean values of ratios for each population were plotted against the time. Error bars represent the relative standard error. Data points from 30–163 min were fitted with an exponential decay function *y*(*x*) = 13.07 exp(−*x*/34.97) +1.90 (*R^2^* = 0.97) resulting in a rate of *k* = (0.029 ± 0.004) min^−1^.

**Figure 6 biosensors-12-00271-f006:**
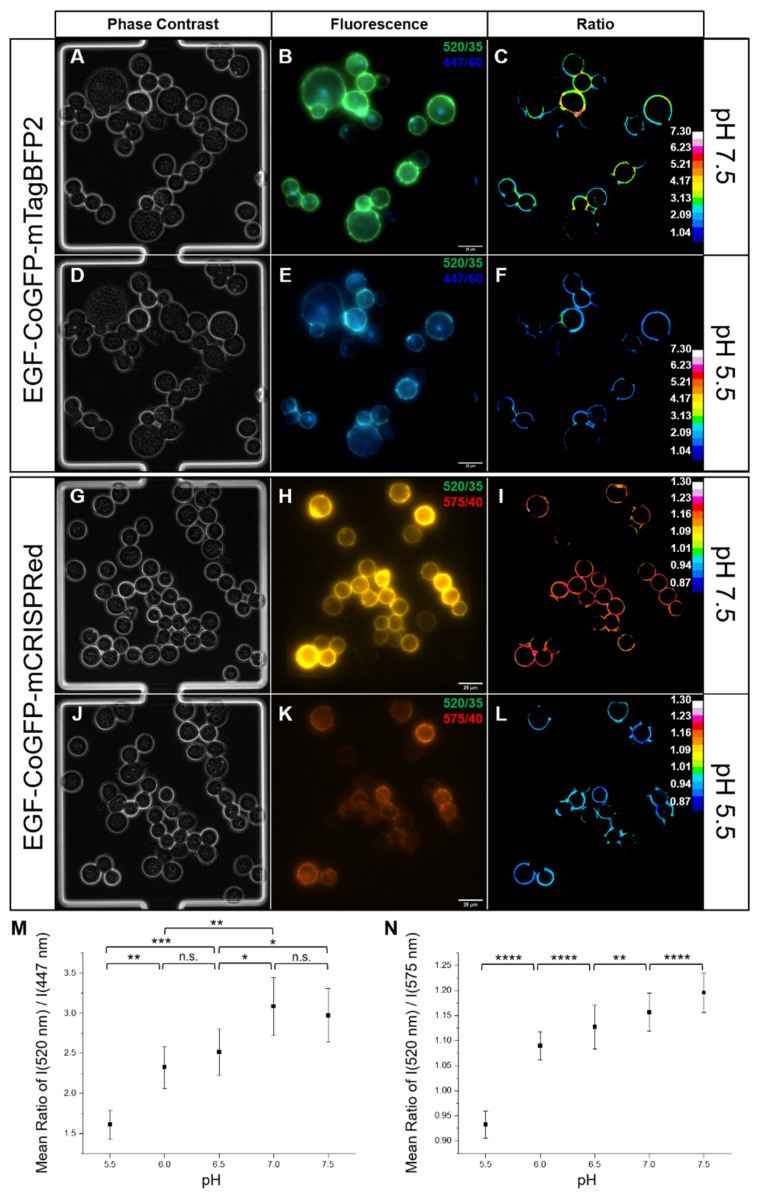
The molecular pH sensors were used in a microfluidic device for single-cell cultivation. EGF-CoGFP-mTagBFP2 (**A**–**F**) or EGF-CoGFP-mCRISPRed (**G**–**L**) bound to 293-F cells expressing the EGFR ectodomain on the cell surface upon incubation. Cells were seeded into the cultivation chambers of the microfluidic device and incubated for 3 h at 22 °C. After 1 h of incubation, cell culture medium at five different pH values was sequentially changed with 5 µL/min and analyzed after 20 min of incubation by fluorescence microscopy. Representative images show the same chambers at pH 7.5 (**A**–**C**,**G**–**I**) and pH 5.5 (**D**–**F**,**J**–**L**) as phase-contrast images (**A**,**D**,**G**,**J**) and overlay fluorescence images of 520/35 and 447/60 nm emission (**B**,**E**) or 520/35 and 575/40 nm emission (**H**,**K**). Pseudo-colored images represent emission ratios 520 nm/447 nm (**C**,**F**) and 520 nm/575 nm (**I**,**L**) calculated from the fluorescence detected solely from the cell surface. For each pH step (5.5, 6.0, 6.5, 7.0, 7.5) the mean ratio among all cells of nine chambers was plotted for EGF-CoGFP-mTagBFP2 ((**M**), *n* = 72–80, ±SEM) and EGF-CoGFP-mCRISPRed ((**N**), *n* = 150–217, ±SD). * *p* < 0.05; ** *p* < 0.01; *** *p* < 0.001; **** *p* < 0.0001, unpaired two-sided *t*-test.

## Data Availability

Not applicable.

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
