# Peer review of "Genetically Encoded Ratiometric pH Sensors for the Measurement of Intra- and Extracellular pH and Internalization Rates"

_biosensors, 2022, doi:10.3390/bios12050271_

Round 1

Reviewer 1 Report

This manuscript by Lennard Karsten and colleagues develops two ratiometric pH-sensitive fluorescent proteins, which can be used as a versatile platform for the detection of intra- and extracellular pH in live cells. Overall, the experiment design was rigorous, and the authors did systematic work and provided fairly concrete data in the manuscript. This reviewer does not have major technically related issues.

Minor comments:

  1. The number of Keywords is too much please revise it to a rational range.
  2. For a better illustration please add a schematic diagram showing the fabrication process of the ratiometric pH biosensors.
  3. Some work also related to fluorescent-based protein sensors for cell biomarker detection could be involved (see Anal. Chem. 2017, 89, 12152; Anal. Chem. 2020, 92, 647).

Author Response

  1. The number of Keywords is too much please revise it to a rational range.

We reduced the number of keywords according to the guidelines.

  1. For a better illustration please add a schematic diagram showing the fabrication process of the ratiometric pH biosensors.

The ratiometric pH biosensors are genetically encoded three-part fusion constructs according to their names EGF-CoGFP-mTagBFP2 and EGF-CoGFP-mCRISPRed. We added a depiction of the linear protein sequence to Fig. 1 with the names of the constituting proteins for a better illustration. If ‘fabrication’ refers to cloning or production of the constructs, we would be happy to clarify this as well. In short, protein expression and purification followed well established procedures for recombinant expression in E. coli as described in the materials and methods section. In the supplement, we provide data of the process. The graphical abstract might also help to set the stage for the experimental data.

  1. Some work also related to fluorescent-based protein sensors for cell biomarker detection could be involved (see Anal. Chem. 2017, 89, 12152; Anal. Chem. 2020, 92, 647).

We added these references in the introduction mentioning synthetic chromophores and fluorescent nanomaterials as a different kind of biosensors. This truly reflects a huge array of biosensors beyond the genetically encoded variants as described in this manuscript.

Reviewer 2 Report

Karsten et al reports two genetically encoded pH sensors for intra and extra cellular pH sensors. The authors report an optimization strategy to derive CoGFP_V0 fluorescent protein. The protein adduct was combined with EGF and pH resistant protein (mTagBFP2 and mCRISPRed). The fluorescent proteins were validated for pH response in solution and in cells. The authors applied the probes in vitro to map out intracellular and extracellular pH. The manuscript is well written and the results are clearly explained. The findings in the manuscript are of general interest to the community of Biosensors, therefore the reviewer recommends acceptance of the article pending some minor changes.   

Minor.

  1. Placing CoGFP_V0 and mTagBFP2 could impact binding of EGF and EGFR. It is suggested to test the binding constant using SPR or similar method.
  2. Additional controls such as non-EGFR cell line could be used to rule out non-specific binding. Similarly, a competition assay with excess amount of EGF might be useful.
  3. Figure S2A. The legend (ex and em) of figure and the graph are different.
  4. It is suggested to validate the fluorescent protein on cells lines with varied EGFR expression or in more acidic pHs (tumor like conditions).

Author Response

  1. Placing CoGFP_V0 and mTagBFP2 could impact binding of EGF and EGFR. It is suggested to test the binding constant using SPR or similar method.

Previously, we reported a KD = 7.6 nM value for the analogue fusion protein EGF-mCherry measured by biolayer interferometry (BLI), which is in agreement with 1.8 nM reported for EGF-EGFR interactions measured by SPR (Feiner et al. 2019). We added flow cytometry data to the supplement (Figure S 11) showing good binding to A431 cells with high EGFR expression level but no binding to MCF7 cells with low level of EGFR, which indicate that the binding of EGF-CoGFP_V0 remained the same as for EGF-mCherry. It is a good suggestion to look again into the binding kinetics. However, in this case we do not expect significant differences upon exchanging the fluorescent protein of the EGF fusion protein and we focused more on characterizing the complex fluorescence properties as well as experiments demonstrating applicability rather than the subtleties of the EGF EGFR interaction.

  1. Additional controls such as non-EGFR cell line could be used to rule out non-specific binding. Similarly, a competition assay with excess amount of EGF might be useful.

Flow cytometry analysis showed no binding of EGF-CoGFP_V0 or EGF-CoGFP-mCRISPRed to MCF7 cells. The additional data were added to the supplementary information as Figure S11. A competition assay for EGF-mCherry with the commercially available hEGF from Gibco as competitor was previously performed and published (Feiner at al. 2019).

  1. Figure S2A. The legend (ex and em) of figure and the graph are different.

The legend in this figure was adjusted for clarification.

  1. It is suggested to validate the fluorescent protein on cells lines with varied EGFR expression or in more acidic pHs (tumor like conditions).

The binding data of the pH sensor were generated with A431, HEK293F with transient EGFR transfection and MCF7 cells (flow cytometry data were added to the supplementary information as Figure S11), which exhibit varying EGFR levels from about 2,000,000 to 2,000 receptors on the cell surface. The pH sensor demonstrates potential for high-throughput in vitro characterization of tumor cell-targeting vehicle proteins for cancer therapy. An acidic tumor microenvironment would clearly affect the ratiometric fluorescence read out of the pH sensor as can be inferred from the presented microfluidic measurements at various pH. We see the detection of endocytosis within tumor microenvironments an advanced topic of high interest, which should be addressed in future research. Culturing representative tumor organoids with stroma cells would pose extra challenges not reflecting the current stage and focus of the project.

Reviewer 3 Report

The authors prepared two genetically encoded ratiometric pH sensors, EGF-CoGFP-mTagBFP2 and EGF- CoGFP-mCRISPRed, for monitoring intra- and extracellular pH and internalization rates. However, the following points should be addressed before accepted.

1.Please check the formatting of the manuscript, there are many tiny mistakes;

2.“Figure 2” was mistakenly written in “Figure S2” in line 405;

3.In Figure 2 A and B, the curves are from pH 4-7, however in Figure 2 C is pH 4 is missing. Please provide this data;

4. The conclusion part is too long to read, please rewrite and summarize it.

Author Response

  1. Please check the formatting of the manuscript, there are many tiny mistakes;

The manuscript was thoroughly checked again for mistakes, which were corrected.

  1. “Figure 2” was mistakenly written in “Figure S2” in line 405;

This mistake was corrected in the manuscript.

  1. In Figure 2 A and B, the curves are from pH 4-7, however in Figure 2 C is pH 4 is missing. Please provide this data;

Figure 2B shows the calculated spectral overlap integrand. “There is no spectral overlap at pH 4” is mentioned in the main text. We clarified this again in the figure caption. Thus, the FRET E calculation for pH 4 in Figure 2C according to Equation (3) was not done as it would not result in a reasonable curve.

  1. The conclusion part is too long to read, please rewrite and summarize it.

This text was shortened.